# Access to stereodefined (*Z*)-allylsilanes and (*Z*)-allylic alcohols via cobalt-catalyzed regioselective hydrosilylation of allenes

Chao Wang [1], Wei Jie Teo[1] & Shaozhong Ge [1]

Hydrosilylation of allenes is the addition of a hydrogen atom and a silyl group to a carbon–carbon double bond of an allene molecule and represents a straightforward and atom-economical approach to prepare synthetically versatile allylsilanes and vinylsilanes. However, this reaction generally produces six possible isomeric organosilanes, and the biggest challenge in developing this reaction is to control both regioselectivity and stereoselectivity. The majorities of the developed allene hydrosilylation reactions show high selectivity towards the production of vinylsilanes or branched allylsilanes. By employing a cobalt catalyst generated from readily available and bench-stable cobalt precursor and phosphine-based ligands, here we show that this reaction proceeds under mild conditions in a regioselective and stereoselective manner, and affords synthetically challenging, but valuable linear *cis*-allylsilanes with excellent stereoselectivity (generally *cis* to *trans* ratios: >98:2). This cobalt-catalyzed (*Z*)-selective allene hydrosilylation provides a general approach to access molecules containing stereodefined (*Z*)-alkene units.

[1] Department of Chemistry, National University of Singapore, 3 Science Drive 3, Singapore 117543, Singapore. Correspondence and requests for materials should be addressed to S.G. (email: chmgsh@nus.edu.sg)

Allylsilanes are a type of organosilanes with an allyl group on a silicon atom, and the stereochemistry around the allylic double bond may be E (cis) or Z (trans). Allylsilanes are synthetically valuable building blocks because of their non-toxicity, high stability and versatile applications in organic synthesis and material science[1–3]. They have been employed as monomers for syntheses of silicon-containing polymers and undergo a variety of organic transformations[4–6]. As such, various methods have been developed to prepare allylsilanes and the majority of these approaches produce thermodynamically more stable (E)-allylsilanes[7–9]. However, the stereoselective synthesis of a wide range of (Z)-allylsilanes still remains challenging and rare[10–14]. Catalytic hydrosilylation of allenes (Fig. 1a) is one of the most straightforward and atom-economical approaches to synthesize (Z)-allylsilanes, provided that catalysts and reaction conditions favoring the formation of (Z)-allylsilanes can be identified.

Allenes can undergo hydrosilylation with hydrosilanes in the presence of transition metal catalysts to produce allylsilanes or vinylsilanes (Fig. 1a)[15–21]. The major difficulty in allene hydrosilylation is to control the regio- and stereoselectivity because multiple vinylsilane and allylsilane products may be generated (Fig. 1a). The majority of metal-catalyzed hydrosilylation of terminal allenes show high selectivity for β,γ-hydrosilylation, affording allylsilanes or vinylsilanes with a terminal alkene group without an issue of the control over Z/E-selectivity (Fig. 1b)[15–20]. However, α,β-hydrosilylation of terminal allenes is more challenging because it can potentially form four isomeric Z/E-allylsilane and Z/E-vinylsilane products (Fig. 1a). Selective formation of one organosilane product out of four possible isomers is of high synthetic importance. Except for a recent example of molybdenum catalyst for α,β-hydrosilylation of allenes with modest Z/E selectivity at 110 °C or under UV irradiation (Fig. 1c)[21], catalysts for selective allene α,β-hydrosilylation that can combine high catalyst activity, high Z-stereoselectivity, broad substrate scope and mild reaction conditions are conspicuously unknown.

Platinum complexes are the most frequently encountered catalysts for hydrosilylation reactions in industry[22–26]. However, there is a growing interest in replacing platinum catalysts with earth-abundant base-metal catalysts for hydrosilylation[27,28]. Accordingly, a tremendous development has been made in cobalt-catalyzed hydrosilylation of alkenes and alkynes[29–43]. The hydrosilylation of allenes has been studied with stoichiometric amounts of the cobalt complex $Co_2(CO)_8$, but this ligandless cobalt catalyst shows low selectivity for α,β-hydrosilylation[44]. A Co-catalyzed allene α,β-hydrosilylation that can selectively produce (Z)-allylsilanes still remains unknown. Driven by our continuous interest in developing base-metal-catalyzed hydrofunctionalization of unsaturated organic molecules, we are interested in identifying a highly Z-selective cobalt catalyst for allene α,β-hydrosilylation. An improved Z/E-selectivity is anticipated for ligated cobalt catalysts due to a greater stereochemical communication between ligand and allene substrate from a relatively smaller cobalt center, comparing with the molybdenum catalyst $Mo(CO)_6$[21]. Herein we report a cobalt-catalyzed stereoselective α,β-hydrosilylation of terminal allenes to prepare (Z)-allylsilanes. Furthermore, we have developed a practical one-pot procedure to access synthetically challenging trisubstituted (Z)-allylic alcohols by combining this cobalt-catalyzed allene hydrosilylation and subsequent oxidation of the resulting (Z)-allylsilanes. In addition, we show that $Co(acac)_2$ can be reduced by $PhSiH_3$ in the presence of bisphosphine ligands to generate well-defined, but catalytically active, Co(I) hydride complexes.

## Results

**Evaluation of reaction conditions.** We initiated our studies of Co-catalyzed hydrosilylation of allenes by evaluating reaction conditions for the reaction of cyclohexylallene with phenylsilane. This reaction can potentially produce six organosilanes from either 1,2-hydrosilylation ((E)-**1a**, (Z)-**1a**, (E)-**1a′**, and (Z)-**1a′**) or 2,3-hydrosilylation (**1a″** and **1a‴**), as depicted in Table 1. We tested this reaction with various cobalt catalysts that were generated in situ from bench-stable $Co(acac)_2$ and phosphine ligands. In general, these experiments were conducted with 2 mol % cobalt catalysts in THF at room temperature for 18 h. The results of these experiments are summarized in Table 1.

The reactions catalyzed by the combination of $Co(acac)_2$ and monophosphine ligands, such as $PPh_3$ and $PCy_3$, proceeded to low conversions of cyclohexylallene and produced a mixture of six products with low selectivities for (Z)-**1a** (Table 1, entries 1 and 2). Improved conversions and selectivities to (Z)-**1a** (83 −91%) were achieved for reactions that were conducted with the combination of $Co(acac)_2$ and bisphosphine ligands, such as dppe, dppp, dppb, dcpe or dppf (Table 1, entries 3–7). In particular, the reaction with the catalyst generated from Co(acac)₂ and rac-binap occurred to full conversion with excellent selectivity (98%) for (Z)-**1a** (Table 1, entry 8). Similarly high

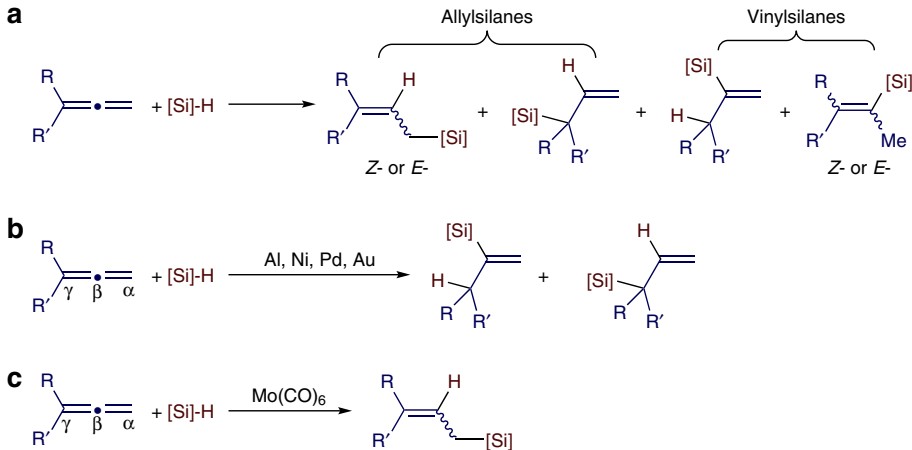

**Fig. 1** Hydrosilylation of allenes. **a** Overview of hydrosilylation of allenes. **b** Metal-catalyzed β,γ-hydrosilylation of allenes. **c** Metal-catalyzed α,β-hydrosilylation of allenes

**Table 1 Evaluation of conditions for cobalt-catalyzed hydrosilylation of cyclohexylallene**

| entry | ligand | solvent | conversion | yield* | product distribution* | | | | | |
|-------|--------|---------|------------|--------|------------|---------|----------|----------|------|-------|
| | | | | | (E)-1a | (Z)-1a | (E)-1a′ | (Z)-1a′ | 1a″ | 1a‴ |
| 1 | PPh₃# | THF | 73% | 65% | 2.6% | 60.0% | 9.7% | 11.8% | 2.0% | 16.9% |
| 2 | PCy₃# | THF | 15% | <10% | 2.9% | 12.2% | 15.0% | 38.6% | 5.7% | 25.7% |
| 3 | dppe | THF | 73% | 51% | 5.6% | 83.6% | 7.1% | 0.9% | – | 2.8% |
| 4 | dppp | THF | >99% | 80% | 10.1% | 83.0% | 2.8% | – | – | 4.1% |
| 5 | dppb | THF | >99% | 79% | 3.3% | 90.9% | 2.0% | – | – | 3.8% |
| 6 | dcpe | THF | >99% | 81% | 5.5% | 88.7% | 4.0% | 0.3% | 0.3% | 1.5% |
| 7 | dppf | THF | >99% | 87% | 2.5% | 91.0% | 3.5% | 1.5% | – | 1.5% |
| **8** | ***rac*-binap** | **THF** | **>99%** | **91%** | **0.7%** | **97.7%** | **1.0%** | **0.1%** | **–** | **0.4%** |
| 9 | xantphos | THF | >99% | 23%♣ | 2.3% | 94.7% | 3.0% | – | – | – |
| 10 | ᵐᵉˢPDI | THF | >99% | 57% | 13.3% | 29.6% | 57.1% | – | – | – |
| 11 | binap | toluene | >94% | 86% | 0.8% | 94.8% | 2.3% | 0.1% | – | 1.4% |
| 12 | binap | hexane | <5 | – | – | – | – | – | – | – |
| 13 | binap | Et₂O | >99% | 91% | 0.8% | 94.6% | 2.6% | 0.2% | – | 1.3% |
| 14 | binap | ᵗBuOMe | >99% | 87% | 0.8% | 94.1% | 2.7% | 0.2% | – | 1.5% |
| 15 | binap | – | 92% | 85%♦ | 0.8% | 92.6% | 3.5% | 0.2% | – | 1.6% |

Reaction conditions: cyclohexylallene (0.500 mmol), PhSiH₃ (0.550 mmol), Co(acac)₂ (10.0 mmol), ligand (10.0 mmol), solvent (1 mL), room temperature, 18 h
Cy cyclohexyl; acac acetylacetonate; THF tetrahydrofuran
*Overall yield of six products and product distribution determined by GC analysis with dodecane as the internal standard
#The loading of ligand was 4 mol%
♣ This reaction afforded bis((Z)-3-cyclohexylallyl)(phenyl)silane **2** in 35% yield
♦ This reaction was conducted on 1.0 mmol scale without any solvent and (Z)-**1a** was isolated in 85% yield

selectivity for (Z)-**1a** was obtained for the reaction catalyzed by Co(acac)₂ and xantphos, but this reaction produced a significant amount of bis((Z)−3-cyclohexylallyl)(phenyl)silane **2** (Fig. 2a), which was generated by the hydrosilylation of cyclohexylallene with (Z)-**1a** as a hydrosilylating reagent (Table 1, entry 9). The result of entry 9 indicates that the hydrosilylation of cyclohexylallene with secondary silanes is likely feasible. Indeed, cyclohexylallene reacted with secondary silanes Ph₂SiH₂ and MePhSiH₂, in the presence of 1 mol% Co(acac)₂ and 1 mol % xantphos, affording the corresponding (Z)-allylsilanes **1b** and **1c** in high yields with excellent stereoselectivities (Fig. 2b, c)[11]. In addition, we tested a nitrogen-based ligand ᵐᵉˢPDI for this transformation, and the reaction afforded a mixture of (Z)-**1a**, (E)-**1a** and (E)-**1a** with low regio- and stereoselectivity (Table 1, entry 10). Furthermore, we also tested various solvents for this hydrosilylation catalyzed by Co(acac)₂/binap (Table 1, entries 11–14). The reactions conducted in toluene, diethylether and *tert*-butylmethylether proceeded to full conversions of cyclohexylallene with excellent selectivity for (Z)-**1a** (Table 1, entries 11, 13 and 14), but the reaction did not occur in hexane, likely due to the poor solubility of the cobalt catalyst in hexane (Table 1, entry 12). In addition, this reaction occurred smoothly in the absence of any solvent and afforded the desired product (Z)-**1a** in 85% isolated yield with Z/E of 99:1 (Table 1, entry 15).

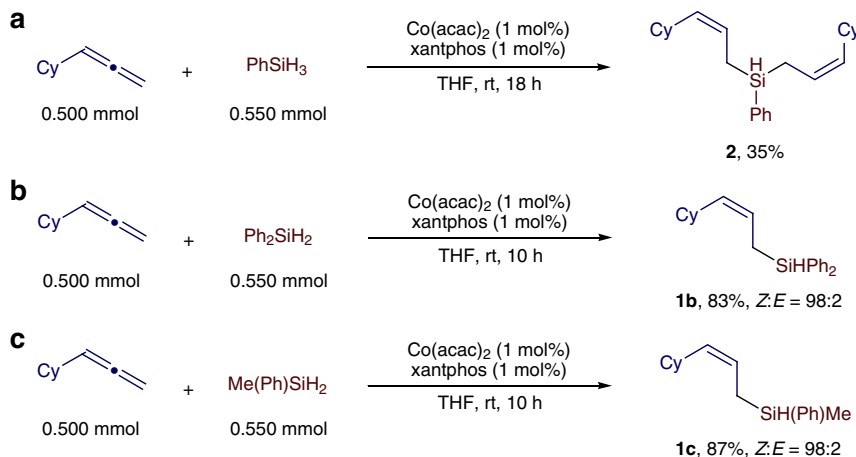

**Fig. 2** Hydrosilylation of cyclohexylallene with secondary silanes. **a** The reaction with phenylsilane PhSiH₃. **b** The reaction with diphenylsilane Ph₂SiH₂. **c** The reaction with methylphenylsilane Me(Ph)SiH₂; Z/E ratios were determined with gas chromatography (GC) analysis on crude reaction mixtures

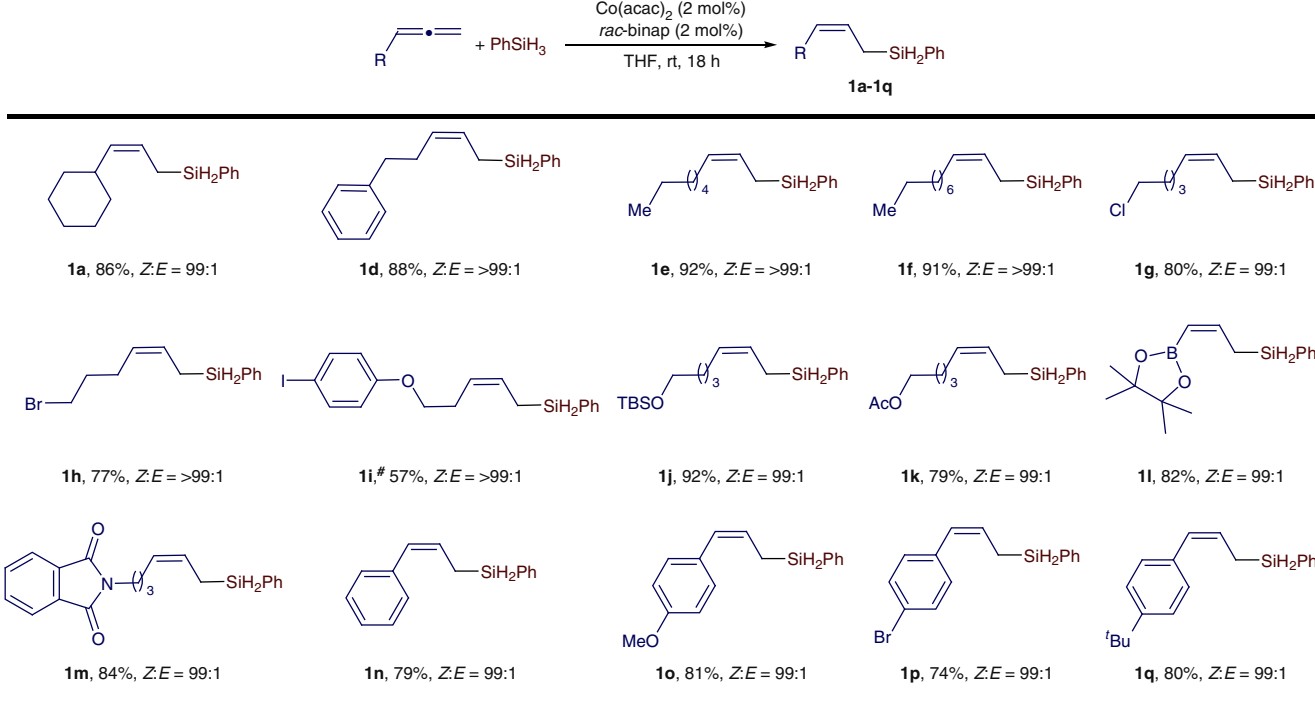

**Fig. 3** Scope of monosubstituted allenes. Conditions: allene (0.500 mmol), PhSiH₃ (0.500 mmol), Co(acac)₂ (10.0 μmol), binap (10.0 μmol), THF (1 mL), room temperature, 18 h; yield of isolated product; Z/E ratios were determined with gas chromatography (GC) analysis on crude reaction mixtures. #3 mol % catalyst, 50 °C

**Substrate scope of allenes**. Under the identified conditions (Table 1, entry 8), we studied the scope of monosubstituted allenes for this reaction. These results are summarized in Fig. 3. In general, a variety of monosubstituted allenes reacted to produce the desired disubstituted (Z)-allylsilanes (1a–1q) in high yields (74–92%) with excellent stereoselectivities (Z:E = 99:1). This reaction shows good functional group tolerance and a range of reactive groups, such as chloro (1g), bromo (1h and 1p), iodo (1i), siloxy (1j), ester (1k), pinacol boronic ester (1l), and imide (1m), are compatible with the reaction conditions. Under the identified conditions, the bromine- and iodine-containing allenes (1h, 1p and 1i) were not fully consumed and this accounts for the relatively lower yields of 1h, 1p and 1i compared with other entries.

Similarly, disubstituted terminal allenes also reacted with PhSiH₃ under the conditions identified for hydrosilylation of monosubstituted allenes (Fig. 3). For example, the reaction between 1-methyl-1-phenylallene and PhSiH₃ proceeded to full conversion in 24 h in the presence of 2 mol% Co(acac)₂ and rac-binap. However, the same reaction conducted with 1 mol% Co(acac)₂/xantphos proceeded to full conversion in only 3 h, affording (Z)-allylsilane 3a in 94% isolated yield with excellent stereoselectivity. Therefore, we chose Co(acac)₂/xantphos to study the scope of disubstituted terminal allenes and the results are

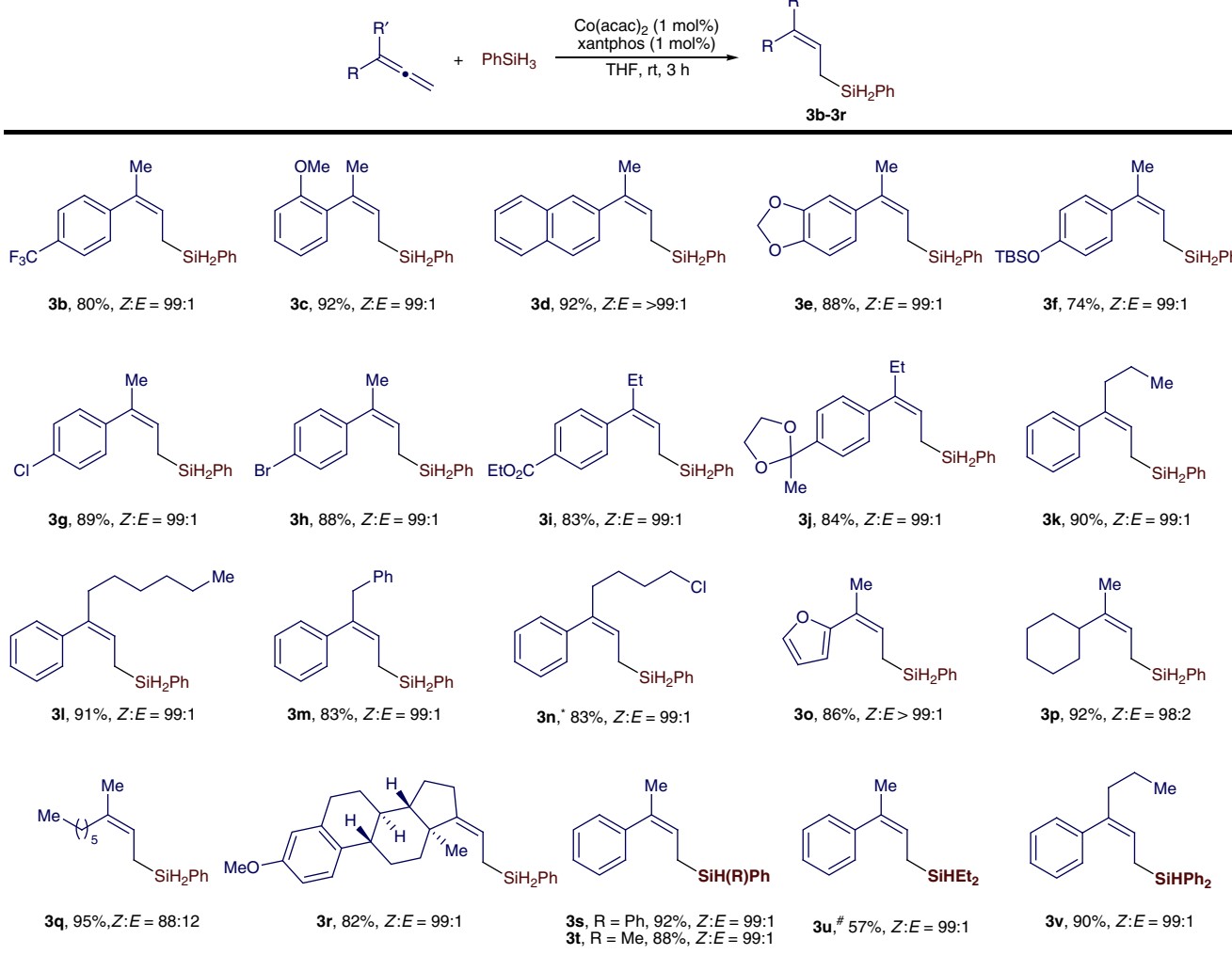

**Fig. 4** Scope of disubstituted allenes. Reaction conditions: allene (0.500 mmol), PhSiH$_3$ (0.500 mmol), Co(acac)$_2$ (5.0 μmol), xantphos (5.0 μmol), THF (1 mL), rt, 3 h; yield of isolated product; Z/E ratios were determined with gas chromatography (GC) analysis on crude reaction mixtures. *2 mol % catalyst, RT, 12 h. #2 mol% catalyst, 70 °C, 24 h

listed in Fig. 4. A series of disubstituted terminal allenes containing aromatic or aliphatic groups readily reacted with PhSiH$_3$ in the presence of 1 mol% Co(acac)$_2$/xantphos at room temperature for 3 h, affording the desired (Z)-allylsilanes (**3b–3s**) in high yields (74–95%) with excellent stereoselectivities (Z:E = 99:1). Decreased Z:E ratio (88:12) was obtained for allene with decreased steric difference between R and R′ (**3q**). A variety of reactive groups, such as siloxy (**3f**), chloro (**3g** and **3n**), bromo (**3h**), ester (**3i**), and acetal (**3j**), are tolerated under the reaction conditions. In addition, this allene hydrosilylation was tested with secondary hydrosilanes (Ph$_2$SiH$_2$, Ph(Me)SiH$_2$, and Et$_2$SiH$_2$), and these reactions produced the corresponding (Z)-allylsilanes (**3s–3v**) in high yields with high stereoselectivities. However, this hydrosilylation of allene did not occur when tertiary hydrosilanes, such as (EtO)$_3$SiH and (Me$_3$SiO)$_2$MeSiH, were used.

**Synthetic potential.** As both Co(acac)$_2$ and the xantphos ligand employed in this transformation are bench-stable, we tested the reaction between 1-methyl-1-phenylallene and PhSiH$_3$ or Ph$_2$SiH$_2$ on a 10 mmol scale in the presence of 0.5 mol% of Co(acac)$_2$ and xantphos that were weighed on the benchtop without using a drybox. These reactions afforded (Z)-allylsilanes **3a** and **3s** in high isolated yield with Z:E of 99:1 (Fig. 5a). In addition, we

demonstrated that (Z)-allylsilane **3v** underwent Ir-catalyzed intramolecular dehydrogenative silylation to afford a six-membered silacyclic compound **4** in 51% isolated yield (Fig. 5b); [dtbpy = 4,4′-di-tert-butyl-2,2′-bipyridyl, nbe = norbornene][45,46].

Allylic alcohols are synthetically valuable intermediates in various organic transformations. The synthesis of stereodefined allylic alcohols, particularly ones containing a trisubstituted Z-alkene unit[47–51], is a persisting challenge in synthetic chemistry. Here we developed a one-pot procedure to synthesize (Z)-allylic alcohols containing disubstituted or trisubstituted alkenes by combining the cobalt-catalyzed allene hydrosilylation and the subsequent oxidation of the resulting (Z)-allylsilanes with H$_2$O$_2$ under basic conditions. A series of (Z)-allylic alcohols (**5a–5h**) can be prepared in good isolated yields using this one-pot procedure (Fig. 5c, see Supplementary Methods for the detailed procedure).

## Discussion

(Z)-allylsilanes are thermodynamically less stable than (E)-allylsilanes and are susceptible to Z/E-isomerization to form the thermodynamically more stable (E)-allylsilanes in the presence of a Co–H species[40,52]. The high Z-selectivities obtained for

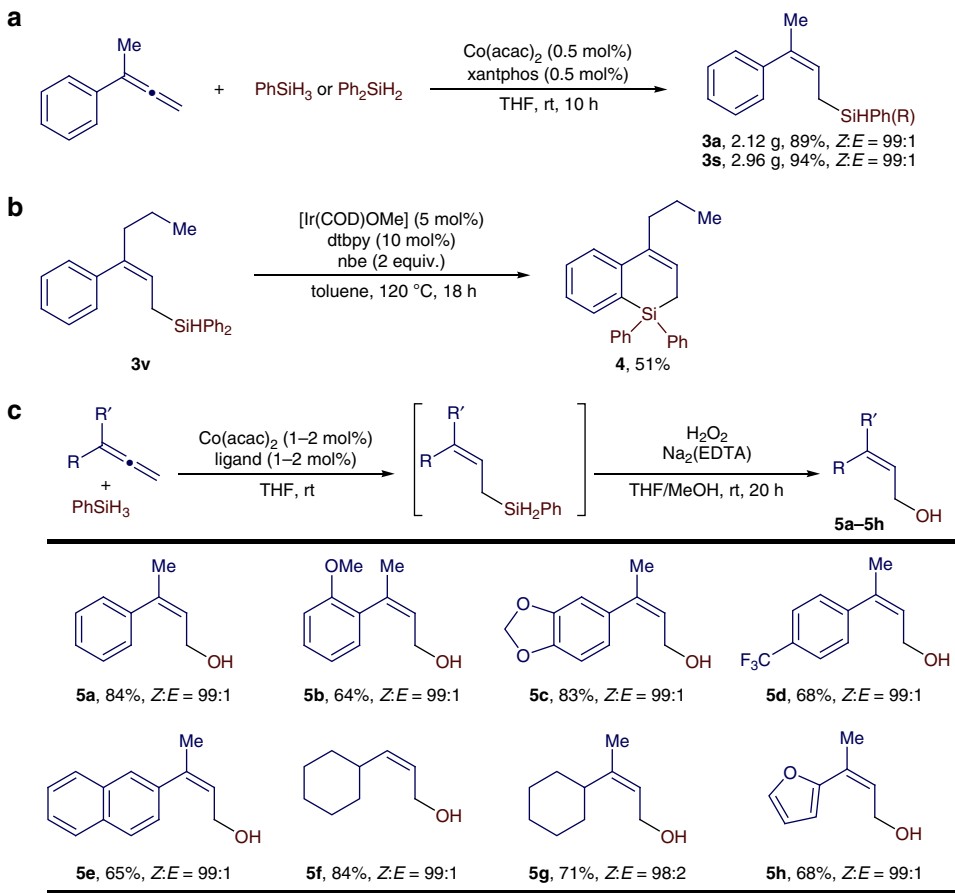

**Fig. 5** Synthetic potential. **a** A concentration of 10 mmol scale reactions. **b** Ir-catalyzed dehydrogenative silylation of (Z)-allylsilane **3v**. **c** One-pot synthesis of (Z)-allylic alcohols; The Z/E ratios were determined with gas chromatography (GC) analysis on crude reaction mixtures

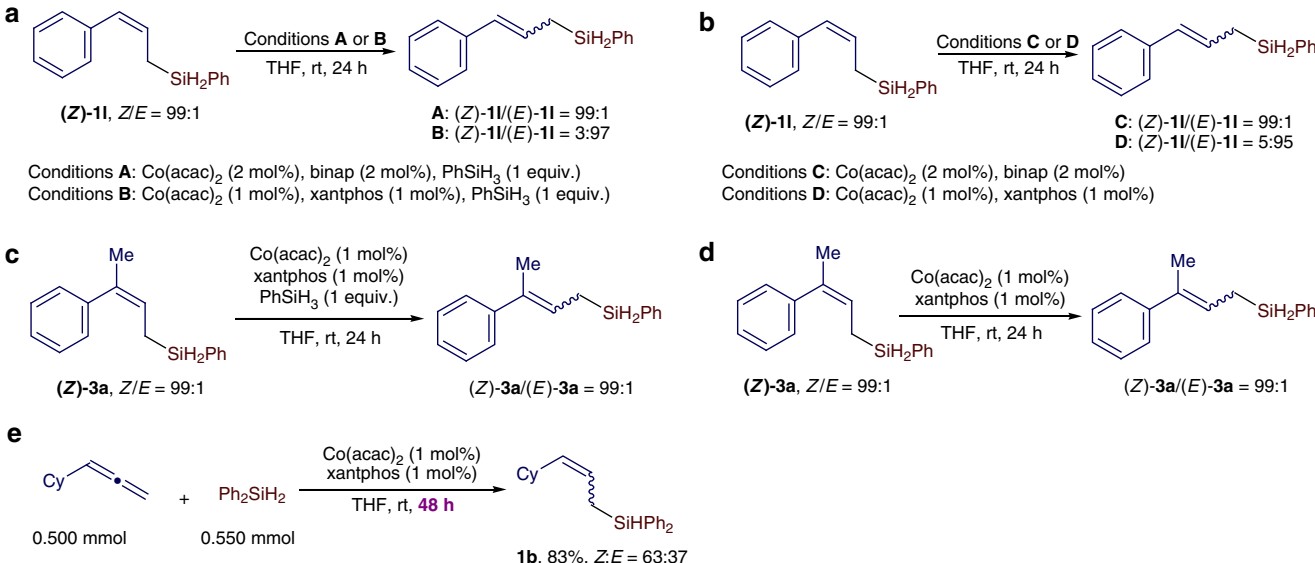

**Fig. 6** Isomerization of (Z)-allylsilanes. **a**, **b** Isomerization of a disubstituted (Z)-allylsilane (Z)-**1l**. **c**, **d** Isomerization of a trisubstituted (Z)-allylsilane (Z)-**3a**. **e** Hydrosilylation of cyclohexylallene with Ph₂SiH₂ conducted for 48 h

reactions listed in Fig. 2 suggest that the cobalt catalyst generated from Co(acac)₂ and binap does not catalyze the Z/E-isomerization of these Z-allylsilanes. Indeed, the isolated (Z)-allylsilane (Z)-**1l** does not undergo Z/E-isomerization in the presence of 2 mol % Co(acac)₂/binap and 1 equivalent of PhSiH₃ (Fig. 6a). However, (Z)-allylsilane (Z)-**1l** was isomerized to (E)-**1l** at room temperature in 24 h in the presence of 1 mol% Co(acac)₂/xantphos and 1 equivalent of PhSiH₃ (Fig. 6a). We also tested these isomerization

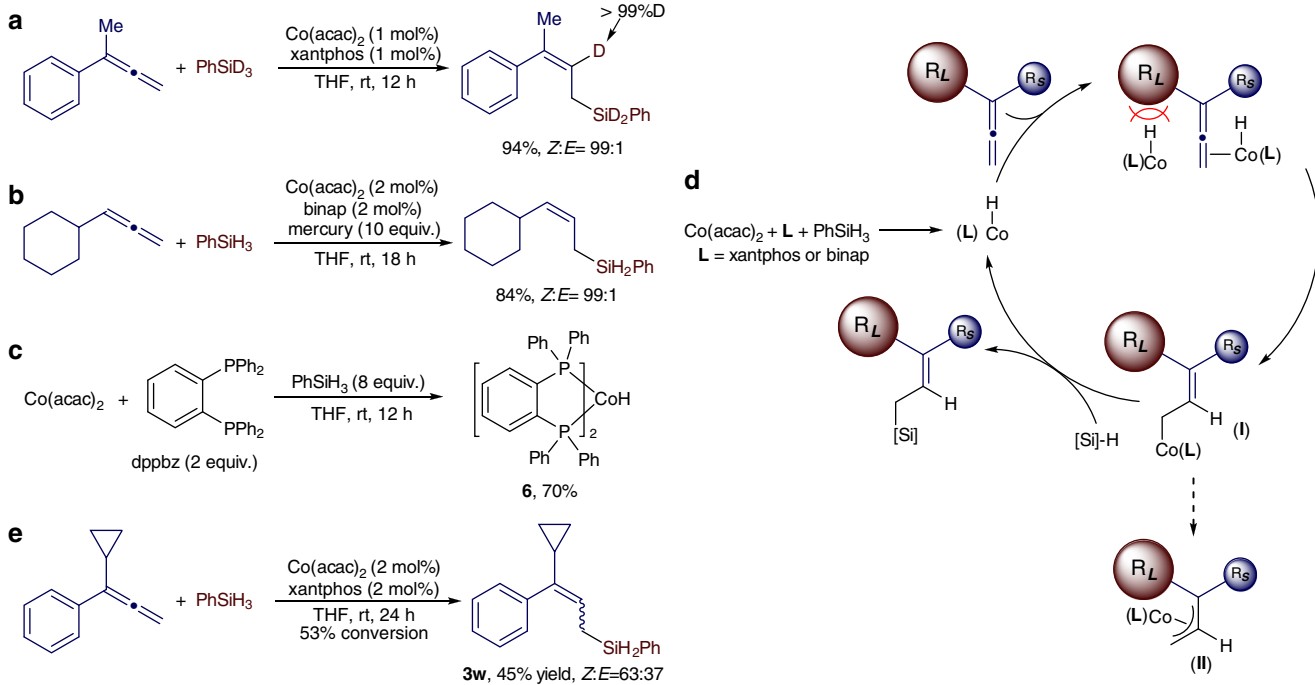

**Fig. 7** Reaction mechanism. **a** A deuterium-labeling reaction. **b** Mercury test. **c** A cobalt hydride complex **6** prepared by reduction of $Co(acac)_2$ with $PhSiH_3$. **d** proposed catalytic cycle for cobalt-catalyzed stereoselective 1,2-hydrosilylation of allenes. **e** hydrosilylation of 1-cyclopropyl-1-phenylallene, an allene containing a radical clock

reactions in the absence of 1 equivalent of $PhSiH_3$ and comparable results were obtained (Fig. 6b). Due to this $Z/E$-isomerization, the reaction in Fig. 2b afforded a mixture of $(Z)$-**1b** and $(E)$-**1b** with a ratio of 63:37 when the reaction time was extended to 48 h (Fig. 6e). Interestingly, the $Z/E$-isomerization of the isolated $(Z)$-**3a** did not occur in the presence of $Co(acac)_2$/xantphos catalyst (Fig. 6c, d). We attributed the lack of $Z/E$-isomerization for the trisubstituted allylsilane $(Z)$-**3a** to the increased steric hindrance around the C=C bond, compared to the disubstituted allylsilane $(Z)$-**1l**.

To understand the mechanism of this cobalt-catalyzed allene hydrosilylation, we conducted a deuterium-labeling reaction between buta-2,3-dien-2-ylbenzene and $PhSiD_3$. This reaction afforded the corresponding $(Z)$-allylsilane with a deuterium atom located *trans* to the phenyl group (Fig. 7a). In addition, a mercury test suggests the homogeneous nature of this catalytic hydrosilylation reaction (Fig. 7b). To provide insight into the cobalt intermediate for this allene hydrosilylation, we tested the reaction of $Co(acac)_2$ and $PhSiH_3$ in the presence of various bisphosphine ligands and found that the reaction using 2 equivalents of dppbz ligand generated a well-defined $Co^I$-H complex $(dppbz)_2CoH$ (**6**) in 70% isolated yield (Fig. 7c)[53]. Complex **6** was active for allene hydrosilylation with regio- and stereoselectivity matching those of the corresponding reaction catalyzed by $Co(acac)_2$ and dppbz ligand (see Supplementary Table 1 for the detailed evaluation of hydrosilylation with complex **6**).

On the basis of the result of the deuterium-labeling experiment (Fig. 7a), the generation of a catalytically active Co(I)–H intermediate (Fig. 7c), and the precedent of the cobalt-catalyzed hydrosilylation of alkenes[27,28], we propose a hydrometalation pathway with a Co(I) hydride intermediate for this Co-catalyzed stereoselective hydrosilylation of allenes (Fig. 7d). In such a mechanism, migratory insertion of the allene substrate into a Co(I)-H species forms an $\eta^1$-bound allylcobalt intermediate (**I**). The steric repulsion between the $R_L$ of the allyl group and the ligand

of the cobalt catalyst makes the formation of $\eta^3$-bound allylcobalt intermediate (**II**) unfavorable. Subsequent $\sigma$-bond metathesis[54] between the $\eta^1$-bound allylcobalt intermediate (**I**) and hydrosilane produces the $(Z)$-allylsilane product and regenerates the Co(I)-H species. Minimizing the steric interaction between the Co(I)-H species and the substituents on the allene substrate accounts for the observed $(Z)$-selectivity, which suggests that reducing the steric difference between two substituents of 1,1-disubstituted allenes will decrease the $Z/E$-selectivity. Indeed, the hydrosilylation of 1-cyclopropyl-1-phenylallene produces a mixture of $(Z/E)$-allylsilanes with a $Z/E$-ratio of 63:37 (Fig. 7e). Such steric interaction has been proposed in a Rh-catalyzed stereoselective hydroformylation of terminal allenes[55].

In summary, we have developed a highly regio- and stereoselective allene hydrosilylation catalyzed by cobalt complexes generated from bench-stable $Co(acac)_2$ and binap or xantphos ligand. A wide range of monosubstituted and disubstituted terminal allenes reacted to afford the corresponding linear $(Z)$-allylsilanes in high yields with excellent stereoselectivities. This cobalt-catalyzed allene hydrosilylation coupled with sequential oxidation of the resulting $(Z)$-allylsilane provided a practical one-pot approach to prepare synthetically challenging $(Z)$-allylic alcohols. Further studies to develop cobalt-catalyzed selective hydrofunctionalization of other types of multiply unsaturated molecules are on going.

## Methods

**General procedure for cobalt-catalyzed allene hydrosilylation**. In an argon-filled dry box, $Co(acac)_2$, phosphine ligand, THF(1 mL) and a magnetic stirring bar were added to a 4-mL screw-capped vial and the mixture was stirred for 5 min. Then terminal allenes (0.500 mmol) and silane (1.1 eq, 0.550 mmol) were added. The vial was sealed with a cap containing a PTFE septum and removed from the dry box. The reaction mixture was stirred at room temperature for 18 h (Fig. 3) or 3 h (Fig. 4) and the resulting solution was concentrated in vacuum. The crude product was purified by column chromatography on silica gel with a mixture of ethyl acetate and hexane as eluent.

**Data availability**. The authors declare that all the data supporting the findings of this study are available within the article and Supplementary Information files, and also are available from the corresponding author upon reasonable request.

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

## Acknowledgements

This work was supported by the grant from NUS Young Investigator Award (R-143-000-630-133) and the Ministry of Education of Singapore (R-143-000-A07-112).

## Author contributions

C.W. planned and conducted most of the experiments; C.W. and W.J.T. prepared substrates for the reaction scope evaluation; S.G. directed the projects and S.G. and C.W. co-wrote the manuscript. All authors contributed to the discussion.

## Additional information

**Competing interests:** The authors declare no competing financial interests.

