## [Peer Review File · Nature Communications]

Reviewer #1 (Remarks to the Author):

The authors developed a Co-catalyzed stereoselective hydrosilylation of terminal allenes to prepare Z-allylsilanes. After extensively screening ligands, the authors eventually found that the Co(acac)₂/BINAP and Co(acac)₂/Xantphos stood out as best catalysts for generation of Z-allylsilanes with high selectivity. The results shown here are good and interesting, however, this reviewer found nothing new in chemistry, no new technology and no new concept or interesting mechanism were found in this manuscript. Moreover, the catalyst and ligands used were not new. The authors have just used the routine screening method to find good ligand, which are extensively used in organic chemistry. Thus, I don't think this manuscript could be published in Nature Communications with good reputations and should be rejected. It might be suitable for publishing in more special journal such as Organic Letters.

Reviewer #2 (Remarks to the Author):

In this manuscript Ge and coworkers demonstrate a highly efficient and selective process for the conversion of mono- and 1,1-disubstituted allenes into linear allyl silanes, with excellent control over both regio- and geometrical control of the olefin. Such control is critical for advancement of synthesis, thus the reaction has the potential to be of significant utility to the community. On this point, the conditions are relatively simple, use commercially available precatalysts and ligands, and can be used outside the glove box and represent a convenient procedure for accessing this allyl silane motif. Thus, I support publication, however, the question of whether this is the appropriate venue is less clear. The primary vertical advancement is the use of the more earth abundant cobalt catalyst rather than a precious metal catalyst. While this result is not entirely surprising, it does represent an important accomplishment. Regardless of where this is published, the authors should try to answer some of the following questions.

Can tertiary silanes be used? Some tertiary silanes are less expensive than those used by the authors, and could potentially lead to increased yields (depending on what the mass balance is for the reactions).

While the functional group tolerance shown in Fig. 3 is good, there are a number of substrates that are not that high yielding. Please comment on the mass balance in these cases, so that the reader can understand what is happening in the reaction.

Fig. 4 shows a good range of substrates, but I suspect that if the alkyl group were a secondary alkyl substituent (i.e. isopropyl) that the selectivity might diminish dramatically. If this is the case, please demonstrate it with an example. People like to know the limitations of chemistry. If it is not the case, please demonstrate this as well.

Presumably the reaction fails when the allene is substituted at both termini. Please demonstrate or comment on this possibility. Obviously, people will be interested to know if you could use R or S BINAP to perform a kinetic resolution.

Many readers will be unfamiliar with the manner in which Co(I)-H behaves. So feel free to elaborate on the fundamental steps. For instance, how is it that Si-H turns over the allyl-Si intermediate?

Supporting information

Jumping right into spectra is an odd approach. Was this intentional?

There appear to be extra alkene signals in spectra 88, 91, and 97, and perhaps more spectra. I am not sure what this signal is from, but if this is an isomer, then it does not seem possible that the selectivities are as high as claimed. Please clarify, and/or clean and adjust yields as needed.

It should be clear how the different allenes are made so that people can assess the utility of this work. Please include synthetic schemes for the allenes and if any changes were made from the references, please include these. Supporting information should make the work easily reproducible

by other chemists.

Reviewer #3 (Remarks to the Author):

The manuscript described an interesting synthetic route to stereo-selectively obtain (Z)-allyl silane. In this methodology, (1) Commercial available base metal catalyst was used. (2) Phosphine ligands were screened. (3) Relatively low catalyst loading and mild conditions were used. (4) High region- and stereo- selectivity was obtained in 18 hours. Function group tolerance and mechanism was also studied in detail. Besides, attempt to carry out reaction in a large scale under dry-box free condition was investigated.

This methodology could be useful in daily organic synthesis to obtain Z-allylsilane, simple catalyst/ligand and mild condition is a plus. And preliminary results from mechanistic study give opportunity to further optimize the catalytic reaction.

Here's some comments:

1. Org. Lett. 2004, 6, 1131 should be included in the examples on Z-allylsilane formation.
2. Though mercury test is not conclusive in some cases, it would be great if mercury test can be carried out to see if the catalytic reaction is homogeneous.
3. Allyl siloxanes are widely used in cross-linked silicone polymer industry. Though alkoxysilane is rarely studied in allene hydrosilylation, I was wondering if the silane like 1,1,1,3,5,5,5-Heptamethyltrisiloxane or (EtO)₃SiH work for this chemistry. If not, what is the reason people rarely report this – simply no reaction?
4. When discussing E/Z isomerization of (Z)-allylsilane, the additional control reaction with no PhSiH₃ should be tested. When there are large amount of PhSiH₃ and allene left in the system, cobalt center prefers binding terminal allene which is much less sterically hindered compared with Z-allylsilane. So when, if any, isomerization happened, there should be no significant amount of PhSiH₃ left in the system.
5. Even though it is less likely that the reaction goes through radical pathways, it would be great to test this hypothesis, especially when Co-alkyl was proposed as key intermediate in the catalytic cycle. Radical clock may be used. Cyclopropylallene can be made easily and may be tested under same reaction condition.
6. Is THF play an important role in this synthesis? Will diethyl ether and tertbutylmethyl ether work for this chemistry? Any solvent effect (steric or polarity)?
7. Can reaction be run under solvent free condition?
8. In the paper, author stated that complex 6 (dppbzCoH) is allene hydrosilylation active, what about the hydrosilylation rate? Hydrosilylation active cannot be used solely to support it is the potential intermediate of the catalytic reaction.
9. In Table 1 foot note. Co and ligand amount should be 10.0 μmol not 10.0 mol.
10. Please suggest in manuscript, how E/Z ratio was determined in Figure 2,3,4,5.
11. In SI, compounds (1d, 1e, 1g, 1h and 1 m) show additional alkene products judging from 1H NMR. Please identify which isomer and how you determine Z/E ratio (was it determined by GC or NMR, I did not see you identify NMR peaks for E isomer for most of your compound).
12. If GC was used to determine E/Z ratio, please give GC parameters and retention time of each isomers.
13. In SI, 1b, 1c, 1f, 1h, 1i, 1k, 1n, 1o, 1p, 1q, 3a, 3b, 3c, 3d, 3e, 3f, 3g, 3h, 3i, 3j, 3k, 3l, 3m, 3n, 3o, 3p, 3q, 3s, 5a, 5b, 5c, 5d, 5e, 5f, 5h, in these NMR either water, silicone grease, hexane, or unidentified impurity was observed. In many cases, even in 13C it shows impurity. Please identify, purify and recalculate yield if necessary.

First of all, we acknowledge these three reviewers for their valuable comments they made to the title manuscript. The point-by-point responses to these comments are detailed *in italic text in blue*.

Reviewer 1

The authors developed a Co-catalyzed stereoselective hydrosilylation of terminal allenes to prepare Z-allylsilanes. After extensively screening ligands, the authors eventually found that the Co(acac)₂/BINAP and Co(acac)₂/Xantphos stood out as best catalysts for generation of Z-allylsilanes with high selectivity. The results shown here are good and interesting, however, this reviewer found nothing new in chemistry, no new technology and no new concept or interesting mechanism were found in this manuscript. Moreover, the catalyst and ligands used were not new. The authors have just used the routine screening method to find good ligand, which are extensively used in organic chemistry. Thus, I don't think this manuscript could be published in Nature Communications with good reputations and should be rejected.

Response: In this manuscript, we developed a highly effective and highly Z-selective hydrosilylation of mono- and disubstituted allenes to prepare Z-allylsilanes using cobalt catalysts generated in situ from a commercially available cobalt precursor Co(acac)₂ and bench-stable phosphine ligands xantphos and binap. The catalysts are operationally simple and are activated by reactions with hydrosilanes, which does not require air- and moisture-sensitive activators, such as Grignard reagents and superhydride NaB(Et)₃H. Due to the lack of general approaches for syntheses of stereodefined Z-alkenyl molecules, this "simple" cobalt-catalyzed protocol represents a practical methodology to access organic molecules containing stereodefined multiply-substituted Z-alkene units.

Reviewer 2

In this manuscript Ge and coworkers demonstrate a highly efficient and selective process for the conversion of mono- and 1,1-disubstituted allenes into linear allyl silanes, with excellent control over both regio- and geometrical control of the olefin. Such control is critical for advancement of synthesis, thus the reaction has the potential to be of significant utility to the community. On this point, the conditions are relatively simple, use commercially available precatalysts and ligands, and can be used outside the glove box and represent a convenient procedure for accessing this allyl silane motif. Thus, I support publication, however, the question of whether this is the appropriate venue is less clear. The primary vertical advancement is the use of the more earth abundant cobalt catalyst rather than a precious metal catalyst. While this result is not entirely surprising, it does represent an important accomplishment. Regardless of where this is published, the authors should try to answer some of the following questions.

1. Can tertiary silanes be used? Some tertiary silanes are less expensive than those used by the authors, and could potentially lead to increased yields (depending on what the mass balance is for the reactions).

Response: We have tested these hydrosilylation reactions with tertiary silanes, but these reactions did not occur. At the moment, these hydrosilylation reactions could be achieved with primary (PhSiH_3) and secondary hydrosilanes (Ph_2SiH_2 , Et_2SiH_2 and PhMeSiH_2). We have added this comment in the revised manuscript. It reads: “However, this hydrosilylation of allene did not occur when tertiary hydrosilanes, such as $(\text{EtO})_3\text{SiH}$ and $(\text{Me}_3\text{SiO})_2\text{MeSiH}$, were used.”

2. While the functional group tolerance shown in Fig. 3 is good, there are a number of substrates that are not that high yielding. Please comment on the mass balance in these cases, so that the reader can understand what is happening in the reaction.

Response: In Fig.3, for examples, the yields of **1h**, **1p**, and **1i** are relatively lower than other substrates. For these three reactions, the full conversions of these bromine- or iodine-containing allenes were not achieved and the relatively lower yields were resulted from relatively lower conversion of allenes. We have added a comment on this matter and it reads: “Under the identified conditions, the bromine- and iodine-containing allenes (**1h**, **1p** and **1i**) were not fully consumed and this accounts for the relatively lower yields of **1h**, **1p** and **1i** compared with other entries.”.

3. Fig. 4 shows a good range of substrates, but I suspect that if the alkyl group were a secondary alkyl substituent (i.e. isopropyl) that the selectivity might diminish dramatically. If this is the case, please demonstrate it with an example. People like to know the limitations of chemistry. If it is not the case, please demonstrate this as well.

Response: We have tested the hydrosilylation of 1-phenyl-1-isopropylallene with PhSiH_3 and this reaction does not occur. However, we conducted the reaction of 1-cyclopropyl-1-phenylallene with PhSiH_3 and found this reaction occurred to 53% conversion. Two products (**Z/E-3w**) with the Z/E ratio of 63:37 were obtained in 45% isolated yield. We have included this example in the revised manuscript.

4. Presumably the reaction fails when the allene is substituted at both termini. Please demonstrate or comment on this possibility. Obviously, people will be interested to know if you could use R or S BINAP to perform a kinetic resolution.

Response: This is a very good comment. We have tested the hydrosilylation of 1-phenyl-3-propylallene with PhSiH_3 in the presence of 5 mol % of $\text{Co}(\text{acac})_2$ and xantphos . This reaction occurred to <10% conversion and three hydrosilylation products with similar ratios were detected by GC-MS. Due to the low conversion of this reaction, we could not confirm the identities of these three products. We

subsequently tested this reaction with BINAP ligand and lower conversion and similar product ratio were achieved. Due to the low conversion and multiple products presented, it is infeasible to test the kinetic resolution on the hydrosilylation of internal allenes.

5. Many readers will be unfamiliar with the manner in which Co(I)-H behaves. So feel free to elaborate on the fundamental steps. For instance, how is it that Si-H turns over the allyl-Si intermediate?

Response: We added the following text to describe the reactivity of Co(I)-H species with relevant references and Fig. 7c. It reads: Migratory insertion of the allene substrate into a Co(I)-H species forms an η^1 -bound allylcobalt intermediate (I). The steric repulsion between the R_L of the allyl group and the ligand of the cobalt catalyst makes the formation of η^3 -bound allylcobalt intermediate (II) unfavourable. Subsequent σ -bond metathesis⁵⁴ between the η^1 -bound allylcobalt intermediate (I) and hydrosilane produces the (*Z*)-allylsilane product and regenerates the Co(I)-H species. Minimizing the steric interaction between the Co^I-H species and the substituents of the allene substrate accounts for the observed (*Z*)-selectivity.

Supporting information

Jumping right into spectra is an odd approach. Was this intentional?

Response: For submissions to Nature Communications, this is the format for the Supplementary Information. It is intentional to put the spectra in the beginning of the Supplementary Information.

There appear to be extra alkene signals in spectra 88, 91, and 97, and perhaps more spectra. I am not sure what this signal is from, but if this is an isomer, then it does not seem possible that the selectivities are as high as claimed. Please clarify, and/or clean and adjust yields as needed.

Response: Spectrum 88 is for **3i**, spectrum 91 is for **3j**, and spectrum 97 is for

3l. For the starting allene substrates to make these three allylsilanes, they contain some amounts of internal alkynes, byproducts for the reactions to make these allene substrates and these internal alkynes and the corresponding allenes are not separable using column chromatography. The additional alkene signals in the spectra 88, 91, and 97 are from the hydrosilylation products of these internal alkynes, but not from the isomers of products from allene hydrosilylation. These products of internal alkyne hydrosilylation are inseparable from the products of allene hydrosilylation. During the revision of the manuscript, we employed other method to prepare the allene, which does not contain the internal alkyne byproduct, for product 3l. In this case, there is no extra alkene signal for product 3l anymore. This clarification is shown in the following scheme and we have added this clarification in the revised manuscript.

When we re-conducted the reactions that afforded **3i** and **3j**, we used a mixture of allene and internal alkyne that contains 0.500 mmol of allene substrate. The yields were calculated based on the calibrated mass of products (with mass of byproduct removed).

It should be clear how the different allenes are made so that people can assess the utility of this work. Please include synthetic schemes for the allenes and if any changes were made from the references, please include these. Supporting information should make the work easily reproducible by other chemists.

Response: In the revised Supporting Information, the schemes for the synthesis of allene starting materials have been included.

Reviewer 3

The manuscript described an interesting synthetic route to stereo-selectively obtain (Z)-allyl silane. In this methodology, (1) Commercial available base metal catalyst was used. (2) Phosphine ligands were screened. (3) Relatively low catalyst loading and mild conditions were used. (4) High region- and stereo-selectivity was obtained in 18 hours. Function group tolerance and mechanism was also studied in detail. Besides, attempt to carry out reaction in a large scale under dry-box free condition was investigated.

This methodology could be useful in daily organic synthesis to obtain Z-allylsilane, simple catalyst/ligand and mild condition is a plus. And preliminary results from mechanistic study give opportunity to further optimize the catalytic reaction.

1. Org. Lett. 2004, 6, 1131 should be included in the examples on Z-allylsilane formation.

Response: This reference has been included in the revised manuscript, as ref. 14. (14. Sawaki, R., Sato, Y. & Mori, M. Ligand-Controlled Highly Stereoselective Syntheses of E- and Z-Allylsilanes from Dienes and Aldehydes Using Nickel Complex. Org. Lett. 6, 1131-1133 (2004).)

2. Though mercury test is not conclusive in some cases, it would be great if mercury test can be carried out to see if the catalytic reaction is homogeneous.

Response: We have tested the hydrosilylation of buta-2,3-dien-2-ylbenzene with PhSiH₃ catalyzed by Co(acac)₂/xantphos (1 mol %) in the presence of 10 equiv. of mercury, and similar results have been achieved compared to the reaction in the absence of mercury. These results indicated the homogeneous nature of this catalytic reaction. This has been included in the revised manuscript and it reads: In addition, mercury test suggests the homogeneous nature of this catalytic hydrosilylation reaction (Fig. 7b).

3. Allyl siloxanes are widely used in cross-linked silicone polymer industry. Though alkoxy silane is rarely studied in allene hydrosilylation, I was wondering if the silane like 1,1,1,3,5,5,5-Heptamethyltrisiloxane or (EtO)₃SiH work for this chemistry. If not, what is the reason people rarely report this – simply no reaction?

Response: With the catalysts generated from Co(acac)₂ and phosphine ligands, the hydrosilylation of allenes with (Me₃SiO)SiMeH or (EtO)₃SiH does not occur. It is likely that these two alkoxy silanes are not reactive enough to activate Co(acac)₂ in the presence of bisphosphine ligands. In addition, we also conducted the hydrosilylation with these two tertiary silanes in the presence of 20 mol % of PhSiH₃ to activate the cobalt catalyst, and these reactions still do not proceed. As such, we conclude that this hydrosilylation does not occur with tertiary hydrosilanes. We have added a comment about this point in the revised

manuscript. It reads: "However, this hydrosilylation of allene did not occur when tertiary hydrosilanes, such as (EtO)₃SiH and (Me₃SiO)₂MeSiH, were used."

4. When discussing E/Z isomerization of (Z)-allylsilane, the additional control reaction with no PhSiH₃ should be tested. When there are large amount of PhSiH₃ and allene left in the system, cobalt center prefers binding terminal allene which is much less sterically hindered compared with Z-allylsilane. So when, if any, isomerization happened, there should be no significant amount of PhSiH₃ left in the system.

Response: We tested the E/Z-isomerization of Z-allylsilane products, listed in Figure 6a and 6b, in the absence of 1 equivalent of PhSiH₃. The results of these isomerization reactions are comparable with the results of the corresponding isomerization reactions with 1 equivalent of PhSiH₃. The results of all these isomerization is shown below and also included in the revised manuscript.

5. Even though it is less likely that the reaction goes through radical pathways, it would be great to test this hypothesis, especially when Co-alkyl was proposed as key intermediate in the catalytic cycle. Radical clock may be used. Cyclopropylallene can be made easily and may be tested under same reaction condition.

Response: We have tested the hydrosilylation of a cyclopropyl-containing allene, 1-cyclopropyl-1-phenylallene and found this reaction occurred to 53% conversion. Two products with the Z/E ratio of 63:37 were obtained in 45% isolated yield. The products from the ring-opening of cyclopropyl ring were not identified. These results suggested that it is unlikely that this catalytic reaction follow a radical pathway. We have included this example in the revised manuscript.

6. Is THF play an important role in this synthesis? Will diethyl ether and tertbutylmethyl ether work for this chemistry? Any solvent effect (steric or polarity)?

Response: THF does not play an important role in this catalytic reaction. Other solvents, such as diethyl ether, tert-butyl methyl ether, toluene, and hexane, have been tested for the hydrosilylation reaction of buta-2,3-dien-2-ylbenzene, and no noticeable solvent effect has been observed. The results of these experiments have been included in the section of Evaluation of Reaction Conditions in the revised manuscript (entries 11-14 in table 1).

7. Can reaction be run under solvent free condition?

Response: We have conducted the reaction of cyclohexylallene (a 1.0-mmol-scale reaction) with PhSiH_3 in the absence of solvents using 1 mol % of $\text{Co}(\text{acac})_2/\text{binap}$ (weighed on the bench-top). This reaction occurred to full conversion of the starting allene and the product **1a** was obtained in high isolated yield. This result has been included in the revised manuscript (entry 15 in Table 1).

8. In the paper, author stated that complex 6 (dppbzCoH) is allene hydrosilylation active, what about the hydrosilylation rate? Hydrosilylation active cannot be used solely to support it is the potential intermediate of the catalytic reaction.

Response: Complex **6** is $(\text{dppbz})_2\text{CoH}$, a 18-electron species. This complex is not an intermediate for this cobalt-catalyzed hydrosilylation. However, this complex can dissociate a phosphine ligand (dppbz) to form a 16-electron species " $(\text{dppbz})\text{CoH}$ " which can undergo migratory insertion with allene substrates. To get the evidence for the ligand dissociation, we tested the hydrosilylation with $\text{Co}(\text{acac})_2$ and various amounts of dppbz ligand and found a strong ligand inhibition effect. Based on this observation, we propose that the 16-electron species " $(\text{dppbz})\text{CoH}$ ", which is not stable enough for isolation, is the catalytically active species for this hydrosilylation reaction. This information has been included in the revised supplementary materials (Supplementary Table 1).

9. In Table 1 foot note. Co and ligand amount should be 10.0 μmol not 10.0 mol.

Response: This error has been corrected in the revised manuscript.

10. Please suggest in manuscript, how E/Z ratio was determined in Figure 2,3,4,5.

Response: In the revised manuscript, we have indicated that the E/Z-ratios were determined by gas chromatography (GC) in Figure 2, 3, 4 and 5. The temperature profile and parameters for the GC column have been included in the revised supplementary materials.

11. In SI, compounds (1d, 1e, 1g, 1h and 1 m) show additional alkene products judging from ¹H NMR. Please identify which isomer and how you determine Z/E ratio (was it determined by GC or NMR, I did not see you identify NMR peaks for E isomer for most of your compound).

Response: For compounds **1d**, **1e**, **1g**, **1h** and **1m**, the additional alkene products from ¹H NMR are not the E-isomers, but the other regioisomers, allylsilanes with a terminal alkene unit (see the picture attached with percentages determined with GC analysis). For most of (Z)-allylsilane products, the (E)-isomers were not present in the crude reaction mixtures as determined from GC-MS analysis and this is the reason that we did not identify NMR peaks for the E-isomers.

12. If GC was used to determine E/Z ratio, please give GC parameters and retention time of each isomers.

Response: The temperature profile and parameters for the GC column, together with the retention time of each isomer, have been included in the revised supplementary materials.

13. In SI, 1b, 1c, 1f, 1h, 1i, 1k, 1n, 1o, 1p, 1q, 3a, 3b, 3c, 3d, 3e, 3f, 3g, 3h, 3i, 3j, 3k, 3l, 3m, 3n, 3o, 3p, 3q, 3s, 5a, 5b, 5c, 5d, 5e, 5f, 5h, in these NMR either water, silicone grease, hexane, or unidentified impurity was observed. In many cases, even in ¹³C it shows impurity. Please identify, purify and recalculate yield if necessary.

Response: For all these listed compounds, we repeated these experiments and purified these compounds with column chromatography with HPLC grade hexane. The isolated yields for these experiments are updated in the revised manuscript.

Reviewer #2:

Remarks to the Author:

In this work, the authors help the field of synthesis take a large step forward in moving toward more sustainable synthesis, by identifying a very robust, earth abundant, catalyst which operates under nearly ideal conditions to give hard to access but very useful products in good to excellent yields and selectivities. This work will certainly influence the direction of the field, even if my sense of "surprise" about the findings is somewhat muted, hence I would support publication. The authors have answered my previous concerns.

Reviewer #3:

Remarks to the Author:

The authors did answer all my questions.

Here I have an additional question - along with the reviewer 2's question on substrate scope. I wanted to ask the same question last time but forgot. The author showed some of the substrates have lower conversion, is it simply because the rate of reaction is low or the substrate is decomposing the catalyst, if so, will higher loading give better conversion, also what catalyst decomposed to?

First of all, we acknowledge the two reviewers for their valuable comments they made to the title manuscript. The point-by-point responses to these comments are listed *in italic text in blue*.

Reviewer #2 (Remarks to the Author):

In this work, the authors help the field of synthesis take a large step forward in moving toward more sustainable synthesis, by identifying a very robust, earth abundant, catalyst which operates under nearly ideal conditions to give hard to access but very useful products in good to excellent yields and selectivities. This work will certainly influence the direction of the field, even if my sense of "surprise" about the findings is somewhat muted, hence I would support publication. The authors have answered my previous concerns.

Response: *This reviewer did not request further revisions.*

Reviewer #3 (Remarks to the Author):

The authors did answer all my questions.

Here I have an additional question - along with the reviewer 2's question on substrate scope. I wanted to ask the same question last time but forgot. The author showed some of the substrates have lower conversion, is it simply because the rate of reaction is low or the substrate is decomposing the catalyst, if so, will higher loading give better conversion, also what catalyst decomposed to?

Response: *This lower conversion might be due to the slower rate of the reaction. For example of the reaction to make **1i** in Figure 3 in the manuscript, almost full conversion could be achieved if the reaction time was elongated to 24 h instead of 18 h.*